# Information Bottleneck-Enhanced Reinforcement Learning for Solving Operation Research Problems

**DOI:** 10.3390/s25247572

**Published:** 2025-12-13

**Authors:** Ruozhang Xi, Yao Ni, Wangyu Wu

**Affiliations:** 1Krieger School of Arts and Sciences, Johns Hopkins University, Washington, DC 20001, USA; rxi4@jh.edu; 2School of Integrated Circuit Engineering, Guangdong University of Technology, Guangzhou 510006, China; 3School of Computer Science, University of Liverpool, Liverpool L69 3DR, UK

**Keywords:** reinforcement learning, information bottleneck, operation research problems

## Abstract

Reinforcement learning (RL) has achieved remarkable success in complex decision-making tasks; however, its application to structured combinatorial optimization problems in operations research (OR) and smart manufacturing remains challenging due to high-dimensional state spaces, inefficient exploration, and unstable training dynamics. In this work, we propose Information Bottleneck-Enhanced Reinforcement Learning (IBE), a novel framework that integrates information-theoretic regularization into attention-based RL architectures to enhance both representation learning and exploration efficiency. IBE introduces two complementary objectives: (1) a state representation bottleneck, which drives the encoder to extract compact and task-relevant representations from high-dimensional sensory or operational data by minimizing redundant information; (2) a policy bottleneck, which regularizes policy optimization through an information-based exploration bonus derived from the mutual information between states and actions. Together, these mechanisms promote more robust representations, smoother policy updates, and more effective exploration in large, structured decision spaces. We evaluate IBE on representative routing and scheduling problems that commonly arise in logistics and sensor-driven manufacturing systems. Experimental results show that IBE consistently outperforms strong RL baselines, including PPO, REINFORCE, AM, and NeuOpt in both performance and stability. Comprehensive ablation studies further confirm the complementary effects of the two bottleneck components. Overall, IBE provides a principled and generalizable framework for improving RL performance in combinatorial optimization and real-world industrial decision-making under Industry 4.0 environments.

## 1. Introduction

Reinforcement learning (RL) has achieved remarkable success in a wide range of complex decision-making domains, from strategic games [1,2,3,4] and robotics [5,6] to, more recently, the post-training of large language models (LLMs) [7,8,9]. These successes highlight the capability of RL to learn effective sequential decision-making strategies directly from interaction.

However, applying RL to real-world problems remains challenging due to the complexity and high dimensionality of observations, and the need for stable and efficient policy learning. In many practical domains, raw observations often contain redundant or irrelevant information, which can distract the policy and slow down convergence. To address this issue, the information bottleneck (IB) principle [10] provides a theoretically grounded approach to learning compact and robust representations. By constraining the amount of information the latent representation retains about the input while preserving the information relevant to the task, IB helps extract the essential features for decision-making and improves generalization and robustness.

Despite extensive progress in RL for control and IB for representation learning, their applications to operations research (OR) problems remains comparatively underexplored. Many OR tasks, such as vehicle routing, delivery scheduling, and resource allocation, involve inherently sequential decision-making under constraints. For instance, a logistics company must plan efficient delivery routes for multiple vehicles to minimize cost and travel time; a ride-hailing platform must dynamically assign and route drivers to serve customers optimally. These problems are typically modeled as combinatorial optimization tasks, often framed as variants of the Traveling Salesman Problem (TSP) or Vehicle Routing Problem (VRP) [11,12]. Solving them efficiently can yield significant improvements in logistics, transportation, and supply chain management [13,14,15,16].

In addition, such routing and scheduling tasks are central to smart manufacturing and industrial automation systems, where sensors and Internet of Things (IoT) devices continuously generate real-time data about production status, machine conditions, and material flows. Leveraging reinforcement learning to process and act upon this sensing data enables intelligent and adaptive decision-making in dynamic industrial environments, thereby improving operational efficiency, flexibility, and sustainability in Industry 4.0 settings.

Recent works on OR have shown that RL-based methods can generalize across problem instances, enabling data-driven solvers that adapt to different task configurations [13,14,15,17]. However, two major challenges persist: (1) robust state representation, as the state space grows combinatorially with problem size and structure; (2) effective exploration, as naive exploration strategies often fail in large and highly structured decision spaces. Addressing these challenges is key to developing RL agents that are both stable during training and generalizable to unseen problem instances.

In this work, we propose Information Bottleneck-Enhanced (IBE) RL method, a novel framework that integrates the information bottleneck principle with attention-based RL architectures to improve both representation learning and exploration in combinatorial optimization tasks. Specifically, IBE introduces two complementary IB objectives:A state representation bottleneck, which encourages the encoder to learn compact and task-relevant representations of the environment by minimizing unnecessary information;A policy bottleneck, which regularizes the policy optimization with an IB-based exploration bonus, encouraging the agent to balance exploitation of known strategies with exploration of informative actions.

Furthermore, we employ attention mechanisms as feature extractors in IBE, leveraging their ability to model complex spatial–temporal dependencies inherent in routing and scheduling problems. This design allows the agent to capture global relational structures among nodes, routes, or time steps, enhancing representation quality and training efficiency. Our key contributions can be summarized as follows:We propose IBE, a new RL framework that integrates dual information bottleneck objectives into both state representation learning and policy optimization. This design improves representation robustness and enhances exploration efficiency in structured decision spaces.We conduct extensive experiments on multiple benchmarks, including two routing tasks and one scheduling problem. Across all settings, IBE consistently outperforms other competitive baselines in terms of performance, generalization, and stability.We perform comprehensive ablation studies to examine the contribution of each component, including the effects of bottleneck strength and the use of the attention encoder. These analyses provide insights into how IB principles improve both learning dynamics and policy generalization.

Overall, IBE bridges the gap between information-theoretic representation learning and reinforcement learning for combinatorial optimization, offering a principled approach to more robust and generalizable RL solutions in operations research domains.

## 2. Related Work and Preliminaries

In this section, we provide a brief overview of related work and introduce the preliminaries necessary to understand this study.

### 2.1. Reinforcement Learning in Operations Research Problems

RL has emerged as a promising paradigm for tackling complex problems in operations research (OR), where traditional optimization algorithms often struggle with scalability, adaptivity, or generalization to dynamic environments. Applications of RL in OR include the Traveling Salesman Problem (TSP) [18], bin and machine packing [19], supply chain management [20], and various Vehicle Routing Problem (VRP) variants [13], among others. Recently, RL has been applied to increasingly complex real-world domains, such as bridge maintenance via multi-agent ranking proximal policy optimization [21] and advanced path planning through ensemble RL [22]. Complementing these practical applications, recent surveys have outlined the field’s emerging challenges and future prospects [23]. These works demonstrate that RL can learn data-driven heuristics capable of generalizing across diverse problem instances, offering a flexible alternative to handcrafted optimization algorithms.

However, most prior approaches focus on task-specific designs and do not pay attention to compact representation learning with IB, where the learning architecture is tightly coupled to a particular OR formulation. In contrast, our work proposes IBE, a general framework that achieves robust performance across multiple OR domains.

#### 2.1.1. Reinforcement Learning Framework

Following the formal definition of a Markov Decision Process (MDP) introduced by Sutton et al. [24], we define an MDP as a 5-tuple:M=(S,A,P,R,γ),
where

S denotes the state space, representing all possible states of the environment;A denotes the action space, consisting of all actions available to the agent;$P(s^{\prime }|s,a)$ defines the state transition probability, i.e., the probability of reaching state $s^{\prime }$ after taking action *a* in state *s*;R(s,a) is the reward function, specifying the scalar feedback received after executing action *a* in state *s*;γ∈[0,1) is the discount factor, controlling the importance of future rewards.

At each discrete time step *t*, the agent observes a state $s_{t}\in S$, selects an action $a_{t}\in A$ according to its policy $\pi (a_{t}|s_{t})$, receives a reward $r_{t}=R\left(s_{t},a_{t}\right)$, and transitions to the next state $s_{t+1}$ sampled from $P(\cdot |s_{t},a_{t})$. The objective of the agent is to learn an optimal policy $\pi^{*}$ that maximizes the expected cumulative discounted reward:$$J\left(\pi \right)=E_{\pi }\left[\sum_{t=0}^{T}\gamma^{t}r_{t}\right],$$
where *T* is the time horizon.

In practice, various RL algorithms are used to optimize this objective. Value-based methods (e.g., DQN [25]) estimate the expected return of state–action pairs, while policy-based methods (e.g., REINFORCE [24]) directly optimize the policy parameters via gradient ascent. Actor–critic methods [26,27] combine both paradigms to improve stability and sample efficiency. The proposed IBE model incorporates IB objectives into actor–critic methods and PPO.

#### 2.1.2. Proximal Policy Optimization

We base our approach on the Proximal Policy Optimization (PPO) algorithm [27], a robust on-policy reinforcement learning method widely adopted for continuous-control and combinatorial optimization tasks. PPO improves stability by constraining the policy update through a clipped surrogate objective:(1)$$L_{\mathrm{PPO}}\left(\theta \right)=E_{t}\left[\min\left(r_{t}\left(\theta \right){\hat{A}}_{t},\;\mathrm{clip}\left(r_{t}\left(\theta \right),1-\epsilon ,1+\epsilon \right){\hat{A}}_{t}\right)\right],$$
where $r_{t}\left(\theta \right)=\frac{\pi_{\theta }\left(a_{t}|s_{t}\right)}{\pi_{\theta_{\mathrm{old}}}\left(a_{t}|s_{t}\right)}$ is the probability ratio between the new and old policies, ${\hat{A}}_{t}$ is the advantage estimate, and ϵ controls the size of the trust region.

### 2.2. Information Bottleneck in Reinforcement Learning

The information bottleneck (IB) principle [10] has been widely adopted in RL to promote compact and task-relevant representations, improving both robustness and generalization. Early studies applied the IB to enhance representation learning in Atari games [28], demonstrating that compressed latent representations can mitigate overfitting to irrelevant visual details. Lu et al. [29] proposed a dynamics-based IB objective to enable rapid adaptation to new environments with changing system dynamics, while Fan and Li [30] extended this idea to temporal and multi-view settings, learning representations robust to task-irrelevant distractions. In the meta-RL context, Xiang et al. [31] integrated variational IB into meta-learning to encourage transferability across tasks, and Igl et al. [32] introduced the Information Bottleneck Actor–Critic (IBAC) framework, which utilizes variational bottlenecks to improve generalization in noisy, low-data regimes.

Despite these advances, most IB-based RL methods primarily target representation robustness or domain generalization in continuous-control or visual domains. Our approach differs by integrating IB principles at two levels in OR problems: (1) a state representation bottleneck, which enforces compressed and informative embeddings; (2) a policy bottleneck, which introduces an IB-inspired regularization term that serves as an exploration bonus. This dual design explicitly connects the IB principle to both representation learning and exploration in combinatorial optimization settings.

#### Information Bottleneck Theory

The information bottleneck (IB) principle provides a theoretical framework for learning compact and task-relevant representations of data. The key idea is to extract a latent variable that retains only the information necessary for predicting the target variable, while discarding irrelevant or redundant details from the input.

Formally, given an input variable *X* and a target variable *Y*, the IB objective seeks a stochastic representation *Z* that maximizes the mutual information I(Z;Y) with the target while minimizing the mutual information I(Z;X) with the input:(2)$$L_{\mathrm{IB}}=I\left(Z;X\right)-\beta I\left(Z;Y\right),$$
where β is a Lagrange multiplier that balances *compression* and *relevance*. A smaller β encourages more information retention from the input, while a larger β enforces stronger compression, leading to more abstract and generalizable representations.

### 2.3. Exploration Bonus in Reinforcement Learning

Exploration remains one of the most fundamental challenges in RL, requiring a balance between exploiting known high-reward actions and exploring new, potentially better strategies. A broad range of methods has been proposed to encourage exploration, such as curiosity-driven rewards [33,34] and diversity-based intrinsic motivation [35], which augment the reward signal with intrinsic bonuses for visiting novel or informative states.

In policy-gradient methods, exploration is often encouraged through entropy regularization, which promotes stochasticity in the policy distribution [27,36]. Entropy-based exploration has proven effective across diverse domains, including robotic control [37], video games [38], and reinforcement learning for large language models [39]. However, its application to operations research problems remains largely unexplored, where the structured and combinatorial nature of decision spaces demands more targeted exploration strategies. In this work, we extend the concept of entropy-based regularization by incorporating an information bottleneck-inspired exploration bonus within our attention-based RL framework. This approach not only encourages more diverse and informative exploration but also leads to consistent performance improvements across routing and scheduling tasks.

### 2.4. Attention Mechanisms in Reinforcement Learning

The attention mechanism has become a cornerstone of modern deep learning, enabling models to selectively focus on relevant parts of the input. Its integration into RL has led to significant advances in both representation quality and decision-making efficiency. The Deep Attention Network (DAN) [40] extends DQN [25] with soft and hard attention modules, allowing the agent to attend to salient regions in visual inputs. Subsequent works have incorporated self-attention within convolutional or recurrent architectures to enhance feature extraction and long-range dependency modeling, yielding notable performance gains in environments such as Atari games [41]. More recently, attention has been explored in various RL settings: Cherepanov et al. [42] proposed a transformer architecture with memory for offline RL in partially observable environments, improving temporal credit assignment; Pramanik et al. [43] investigated efficient recurrent self-attention mechanisms to reduce computational cost while retaining the ability to capture long-term dependencies; and Hu et al. [44] provided a comprehensive overview of attention in multi-agent reinforcement learning (MARL) [45], highlighting its role in communication and coordination among agents. In this work, we leverage attention as a representation backbone for combinatorial optimization tasks, where spatial and temporal dependencies are central. By combining attention with the information bottleneck principle, our IBE framework achieves robust feature extraction and effective exploration in large, structured decision spaces.

#### Attention as a Feature Encoder

We employ an attention mechanism [14] as the feature encoder to extract rich and task-relevant information from the input state. Given an input state represented as feature vectors(3)$$S=\left[s_{1},s_{2},\ldots ,s_{N}\right],\;s_{i}\in R^{d},$$
the self-attention mechanism projects these features into queries (*Q*), keys (*K*), and values (*V*),(4)$$Q=SW_{Q},\;K=SW_{K},\;V=SW_{V},$$
and computes contextualized embeddings as(5)$$\mathrm{Attention}\left(Q,K,V\right)=\mathrm{softmax}\;\left(\frac{QK^{⊤}}{\sqrt{d_{k}}}\right)V.$$

This allows each element to attend to all others, integrating both local and global context. Compared to fully connected encoders, attention provides several advantages for RL: (1) permutation invariance, enabling the model to handle unordered sets; (2) global reasoning, as each entity’s representation depends on all others; (3) scalability and stability, due to parallel computation and effective gradient flow.

In the IBE framework, the attention encoder produces latent representations $T=f_{\theta }\left(S\right)$ that feed into the information bottleneck module.

## 3. Problem Statement

In this section, following the formalization of previous work [46] and definitions in Section 2, we describe how we formalize three tasks used in this work as MDPs, which can be tackled by reinforcement learning techniques. We consider three different tasks, two routing tasks and one scheduling task. Illustrations of all three tasks can be seen in Figure 1.

### 3.1. Task 1: Traveling Salesman Problem

As shown in the left sub-figure of Figure 1, at each decision step, the agent selects the next city to visit. The reward remains zero until all cities have been visited. Once every city has been visited, the reward is defined as the negative total path length. Consequently, maximizing the reward is equivalent to minimizing the total travel distance.

#### 3.1.1. State Space

The state space of the TSP task can be found in Table 1.

#### 3.1.2. Action Space

At each decision step, the agent selects the next city to visit from the set of unvisited cities. The action space is therefore discrete and defined asA={0,1,2,…,N−1},
where *N* is the total number of cities. Each action $a_{t}\in A$ corresponds to the index of the next city to visit.

#### 3.1.3. Reward Function

The reward function is designed to encourage shorter tours by assigning a terminal reward equal to the negative total tour length. During intermediate steps, the agent receives zero reward until the tour is complete. Maximizing the cumulative reward is therefore equivalent to minimizing the total tour length.

### 3.2. Task 2: Split-Delivery Vehicle Routing Problem

The Split-Delivery Vehicle Routing Problem (SDVRP) is a generalization of the Capacitated Vehicle Routing Problem (CVRP), in which customers can be visited multiple times and their demand may be fulfilled partially during each visit. As shown in the middle sub-figure in Figure 1, at each step, the agent selects the next customer to visit based on its current location and remaining vehicle capacity. Upon visiting a customer, the remaining capacity is updated accordingly. If the remaining capacity is insufficient to serve any remaining customers, the agent must return to the depot to reload. The agent receives a reward of 0 unless all customers are fully served. Once all demands are met, the reward equals the negative total route length. Thus, maximizing the reward is equivalent to minimizing the total travel distance.

#### 3.2.1. State Space

The state space of the SDVRP task can be found in Table 2.

#### 3.2.2. Action Space

At each decision step *t*, the agent selects an action $a_{t}\in \left\{0,1,2,\ldots ,N\right\}$, where 0 represents returning to the depot and 1,…,N correspond to customer indices. Only feasible actions (i.e., customers whose remaining demand can be partially or fully served given the remaining capacity, or the depot when capacity is exhausted) are allowed. After executing an action, the environment updates the current location, adjusts the remaining demand and vehicle capacity, and recalculates the action mask accordingly.

#### 3.2.3. Reward Function

The environment provides a sparse, terminal reward that encourages the agent to minimize the total travel distance while satisfying all customer demands. Maximizing the cumulative reward is equivalent to minimizing the total travel distance while fully satisfying all customer demands.

### 3.3. Task 3: Single-Machine Total Weighted Tardiness Problem

The SMTWTP involves scheduling a set of jobs on a single machine. As shown in the right sub-figure of Figure 1, each job *i* is characterized by a processing time, a weight, and a due date. The goal is to minimize the total weighted tardiness, defined as the sum over all jobs of the product between each job’s weight and its tardiness—where tardiness is the amount by which a job’s completion time exceeds its due date (zero if completed on time). In a reinforcement learning formulation, the agent sequentially selects which job to process next. The agent receives a reward of 0 for all intermediate steps and a final reward equal to the negative of the total weighted tardiness once all jobs are completed. Thus, maximizing the cumulative reward corresponds to minimizing the total weighted tardiness objective.

#### 3.3.1. State Space

The state space of the SMTWTP task can be found in Table 3.

#### 3.3.2. Action Space

At each decision step *t*, the agent selects one job $a_{t}\in \left\{1,2,\ldots ,N\right\}$ to process next on the single machine. A dummy starting node (indexed as 0) is masked out and never chosen. After each selection, the chosen job is marked as processed in the action mask,$$m_{t+1}^{a_{t}}=0,$$
and the current processing time is increased by the selected job’s processing duration. The episode terminates once all jobs have been scheduled.

#### 3.3.3. Reward Function

The SMTWTP environment provides a sparse terminal reward that encourages minimizing the total weighted tardiness of all jobs. During intermediate steps, the agent receives no reward while selecting the next job to process. After all jobs are completed, the environment calculates the total weighted tardiness based on how late each job finishes relative to its due date, weighted by its importance. The final reward is the negative of this value, so higher rewards correspond to schedules with lower overall tardiness. Maximizing the reward therefore leads the agent to prioritize jobs efficiently to minimize delays and meet due dates.

## 4. Proposed Method: Information Bottleneck-Enhanced RL (IBE)

### 4.1. Overview and Motivation

IBE introduces two additional information-theoretic regularizers to the standard reinforcement learning objective (in our work, we employ PPO). Specifically, IBE incorporates (1) a state representation bottleneck that encourages the encoder to extract compact, task-relevant representations while discarding irrelevant details; (2) a policy bottleneck that promotes more diverse and balanced exploration.

### 4.2. State Representation Bottleneck (Contribution 1)

Let *S* denote the input state and $T=f_{\theta }\left(S\right)$ the latent embedding produced by the attention encoder. We model the encoder as a stochastic mapping(6)$$T=f_{\theta }\left(S\right)+\varepsilon ,\;\varepsilon ∼N\left(0,\sigma^{2}I\right),$$
which allows control over the amount of information transmitted from *S* to *T*. In all experiments, we set σ = 0.1. This value follows standard practice in variational information bottleneck implementations and produces stable behavior across tasks. According to the IB principle [10], we aim to maximize the expected return while minimizing the mutual information between *S* and *T*:(7)$$\max_{\theta }\;E_{\pi_{\theta }}\left[R\right]-\beta_{1}I\left(T;S\right).$$

In practice, the mutual information I(T;S) is approximated by the KL divergence between the encoder distribution and a prior p(T), leading to the following regularization term:(8)$$L_{\mathrm{state}}=\beta_{1}D_{\mathrm{KL}}\;\left(p_{\theta }\left(T|S\right)\;∥\;p\left(T\right)\right),$$
where p(T) is a Gaussian prior N(0,I) and $\beta_{1}$ controls the strength of state compression. The Gaussian prior represents a maximally entropic distribution under fixed variance, corresponding to a neutral “uninformative” latent space that encourages compact, noise-robust state representations. The KL term $D_{\mathrm{KL}}\left(p_{\theta }\left(T|S\right)\|p\left(T\right)\right)$ serves as a variational upper bound on the mutual information I(T;S), consistent with the variational information bottleneck framework. While this bound may be loose when the true marginal distribution of *T* deviates significantly from the Gaussian prior p(T), it provides a tractable and stable surrogate objective that effectively controls the representation compression in practice, which is crucial for high-dimensional OR problems.

### 4.3. Policy Bottleneck for Exploration (Contribution 2)

Similarly, we introduce a policy bottleneck that constrains the mutual information between states and actions:(9)$$L_{\mathrm{policy}}=\beta_{2}D_{\mathrm{KL}}\;\left(\pi_{\theta }\left(A|S\right)\;∥\;p\left(A\right)\right),$$
where p(A) is a uniform prior distribution over the action space. If p(A) is uniform, this KL divergence simplifies to $D_{\mathrm{KL}}\;\left(\pi_{\theta }\left(A|S\right)∥p\left(A\right)\right)=-H\left(\pi_{\theta }\left(A|S\right)\right)+\mathrm{const}$, which recovers the standard entropy regularization term. Hence, the policy bottleneck generalizes entropy-based exploration by offering a principled information-theoretic interpretation: actions are encouraged to remain as uninformative about the state as possible while still maximizing expected return. Crucially, this formulation exposes the prior p(A) as a flexible design parameter. Unlike standard entropy bonuses, which are rigidly tied to a uniform target, our framework allows p(A) to be tailored to the problem structure. This flexibility enables the injection of domain knowledge directly into the regularization term, anchoring the agent to plausible solutions while still allowing deviation when profitable. For example, in a vehicle routing problem, a non-uniform prior p(A) could be designed to assign higher probability to nearest-neighbor transitions or routes that respect time windows. This effectively ‘soft-starts’ the exploration process, preventing the agent from wasting computational resources on obviously invalid solutions while preserving the stochasticity needed to escape local optima.

### 4.4. Integration with PPO and Training Algorithm (Contribution 3)

By combining the two bottleneck regularizers with the PPO surrogate loss (see Equation (Equation 1)), the overall optimization objective is given by(10)$$L_{\mathrm{IBE}}\left(\theta \right)=L_{\mathrm{PPO}}\left(\theta \right)-\beta_{1}D_{\mathrm{KL}}\;\left(p_{\theta }\left(T|S\right)\;∥\;p\left(T\right)\right)-\beta_{2}D_{\mathrm{KL}}\;\left(\pi_{\theta }\left(A|S\right)\;∥\;p\left(A\right)\right).$$

Here, $\beta_{1}$ controls the degree of compression in the attention encoder and $\beta_{2}$ regulates the exploration–exploitation balance in the policy. When $\beta_{1}=0$ and $\beta_{2}=0$ are uniform, the objective reduces to the original PPO. We set $\beta_{1}=\beta_{2}=0.01$ for all experiments, unless otherwise specified.

The pseudo-code of IBE is presented in Algorithm 1. Empirically, the coefficients $\beta_{1}$ and $\beta_{2}$ can be tuned to balance compression and exploration: smaller values recover standard PPO behavior, whereas larger values encourage more compact latent representations and smoother, more exploratory policies.

To handle the masked actions in routing and scheduling problems, the prior p(A) is defined as a dynamic uniform distribution over the set of feasible actions at step *t*. Specifically, $p\left(a\right)=1/|A_{t}|$ if action *a* is valid, and 0 otherwise. During optimization, we calculate the KL divergence strictly over the valid action space. This is implemented by masking the logits of invalid actions prior to the softmax operation, ensuring the policy assigns zero probability to infeasible moves.
**Algorithm 1:** Information Bottleneck-Enhanced (IBE) PPO
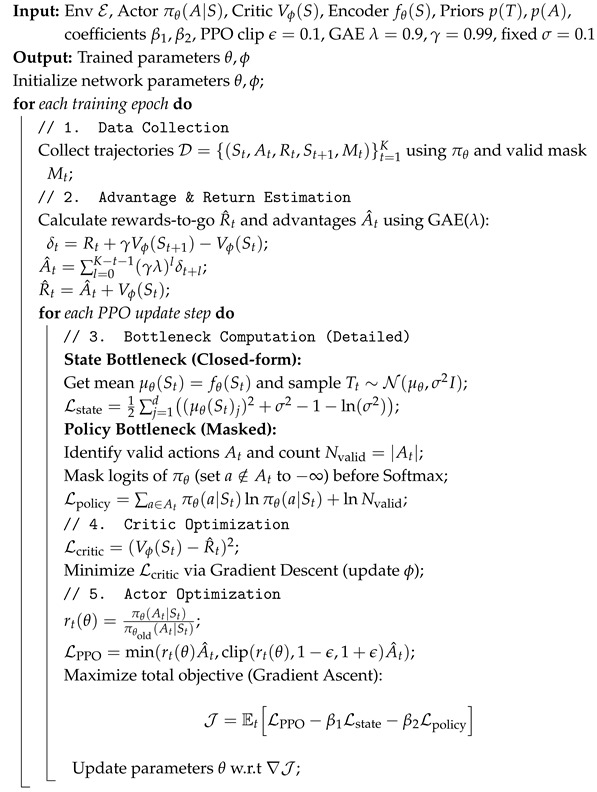


## 5. Experiments

In this section, we empirically evaluate the proposed IBE framework and compare it with six baseline methods: PPO, REINFORCE, Nearest Neighbor, AM, NeuOpt, and a random policy. We begin by describing the experimental environments and setup, followed by quantitative evaluations and ablation studies that assess the contribution of each component within IBE.

Through these experiments, we aim to address the following key research questions:How does IBE perform on routing and scheduling tasks compared with standard reinforcement learning baselines?What is the individual contribution of each component, particularly the two information bottleneck objectives, to the overall performance of IBE?What does the learned policy of IBE look like, and how does it evolve during training?

### 5.1. Environments

We evaluate IBE on three benchmark environments introduced in Section 3: the Traveling Salesman Problem (TSP), the Split-Delivery Vehicle Routing Problem (SDVRP), and the Single-Machine Total Weighted Tardiness Problem (SMTWTP). These environments represent canonical tasks in routing and scheduling, allowing us to assess the generality and scalability of the proposed approach.

For the TSP environment, we generated problem instances with three different scales, where the number of cities (nodes) was set to 50, 100, and 200. In the SDVRP environment, we considered three scales, corresponding to 20, 50, and 100 customers. For the SMTWTP environment, we evaluated instances with 10, 20, and 50 jobs.

### 5.2. Baselines

We compare the proposed IBE framework against four RL representative baselines:PPO [27]: A widely used on-policy reinforcement learning algorithm known for its stability and strong empirical performance across various domains.REINFORCE [24]: A classic policy-gradient method that has been applied successfully to routing and scheduling problems in prior work.AM [14]: A state-of-the-art RL method for solving OR problems which combines an attention mechanism with REINFORCE.NeuOpt [47]: A RL method that utilizes a dual-aspect collaborative transformer to iteratively improve solutions.Nearest Neighbor (NN): A non-learning method for solving routing problems, representing partial solutions as paths with defined start and end nodes. Since it does not involve a learning process, we omit learning curves and report only its final performance in Table 4.Random: A baseline agent that selects actions uniformly at random, serving as a reference for assessing the relative performance improvements achieved by learning-based methods.

### 5.3. Implementation Details

We used a learning rate = 1 × 10^−4^, batch size = 512, and the `Adam’ optimizer. In each epoch, we sampled data and performed gradient updates until all data was sampled. In total, we trained for 10 epochs. In GAE, we set λ=0.9 and γ=0.99. For the attention encoder, we used four attention heads and three layers, with 128-D embedding and a ReLU activation function. We did not use any normalization layers within the transformer blocks. For the fully connected (FC) layers used in the policy/value heads, we used the same 128 embedding dimensions; different numbers of layers were examined in Table 5. All non-attention-based baselines use the same underlying fully connected encoder architecture (three FC layers with 128-D embedding), and AM uses the same attention encoder (three layers, four heads, 128-D embedding) as the IBE framework for fair comparison. The hyper-parameters for all baselines are consistent with the ones used for our method.

### 5.4. Performance in the TSP Environment

We first evaluate the performance of IBE and baseline methods in the TSP environment. The comparative results across different problem scales are shown in Figure 2. Overall, IBE consistently outperforms PPO, REINFORCE, and Random in both convergence speed and final solution quality, demonstrating the effectiveness of the information bottleneck design for representation learning and exploration.

When the task scale is relatively small (N=50), all five agents—IBE, PPO, REINFORCE, AM, and NeuOpt—are able to learn reasonable solutions. However, IBE exhibits significantly faster convergence, achieving near-optimal performance within a fraction of the training epochs required by other baselines. While AM eventually converges to a similar performance level as IBE, it requires more training epochs to do so.

As the problem size increases (N=100), the performance gap between IBE and the baselines widens. All baseline methods show slower learning and converge to suboptimal performance, whereas IBE maintains a smooth and steady improvement curve, ultimately achieving notably higher rewards. This result highlights IBE’s superior scalability and sample efficiency as problem complexity grows. For large-scale tasks (N=200), PPO and REINFORCE struggle to maintain stable training, often exhibiting sharp performance fluctuations or even sudden collapses. With attention, AM is quite stable but learns slowly, possibly attributable to the absence of IB objectives. In contrast, IBE demonstrates much greater training stability and robustness, attributed to the dual information bottleneck objectives that regularize both state representations and policy updates. These results suggest that the proposed IBE method effectively improves policy consistency and performance.

We further visualize the learned policies in Figure 3, which depicts the agents’ trajectories at the beginning and end of training for two different TSP instances. Before training, the paths generated by the IBE agent are random and unstructured, reflecting the agent’s lack of prior knowledge. After 10 training epochs, the agent learns to construct clear and efficient routes that closely approximate optimal solutions.

### 5.5. Performance in the SDVRP Environment

We further evaluate IBE and baseline methods in the SDVRP environment. The quantitative results across different task scales are presented in Figure 4, and corresponding trajectory visualizations are shown in Figure 5. Overall, IBE consistently achieves superior performance, demonstrating faster convergence and greater stability compared to PPO, REINFORCE, and AM.

For the smaller-scale problems (N=20 and N=50), all methods successfully learn feasible routing strategies that outperform the random agent. However, IBE quickly reaches near-optimal performance within the first few training epochs, while PPO, REINFORCE, and AM improve more gradually and plateau at lower reward levels. This result suggests that the information bottleneck helps IBE capture the essential structural patterns of the routing problem more efficiently.

In the large-scale SDVRP setting (N=100), PPO shows unstable training and exhibits severe performance degradation. In contrast, IBE continues to improve steadily, achieving a significant margin over all baselines.

The qualitative visualizations of IBE’s solutions can be seen in Figure 5. At the start of training (green box), the routes produced by the IBE agent are random and inefficient, with overlapping paths and redundant deliveries. After training (blue box), the agent learns to construct well-organized delivery routes that minimize travel distance and effectively split deliveries among vehicles. The clear evolution from disordered to structured trajectories demonstrates IBE’s ability to discover efficient routing strategies.

### 5.6. Performance in the SMTWTP Environment

Finally, we evaluate IBE and the baseline methods on the SMTWTP task, a classic scheduling benchmark. The results for different problem scales are shown in Figure 6. Consistent with our previous findings on routing tasks, IBE outperforms all baseline methods across all SMTWTP instances, though the performance margins are relatively smaller.

For the smaller-scale problems (N=10 and N=20), IBE, REINFORCE, and AM achieve comparable final performance, indicating that REINFORCE is capable of learning reasonable scheduling strategies in simpler settings (AM also uses REINFORCE as its underlying learning algorithm). However, IBE converges much faster, reaching near-optimal performance within the early training epochs. PPO, on the other hand, learns significantly slower and converges to a suboptimal solution.

As the task size increases (N=200), the performance differences become more evident. IBE continues to improve smoothly and maintains high stability throughout training, while PPO exhibits strong fluctuations and often fails to consistently improve. Interestingly, REINFORCE performs competitively in this setting, achieving similar final performance to IBE but with slightly slower convergence. This suggests that the IB objectives primarily enhances training efficiency and stability in smaller scheduling problems.

Ffinal performance results and computation times are shown in Table 4. Overall, these results confirm that IBE generalizes well beyond routing tasks and remains robust across different combinatorial optimization settings. Its faster convergence and smoother learning dynamics demonstrate the effectiveness of incorporating information-theoretic regularization. However, IBE requires more computation time than other methods, as leveraging PPO for updates and employing an attention-based encoder introduces additional computational overhead.

**Table 4 sensors-25-07572-t004:** Final performance (R is short for reward) and computation time (T in minutes) comparison across methods on different tasks, illustrated with standard deviation (±std) values. Results are averaged over 3 independent runs.

**Task**	**N**	**IBE**		**PPO**		**REINFORCE**	
		R (±std)	T (±std)	R (±std)	T (±std)	R (±std)	T (±std)
**TSP**	50	**−5.5** ± 0.58	32 ± 6.2	−6 ± 0.96	29 ± 4.1	−6 ± 0.44	6 ± 0.44
	100	**−9** ± 0.48	58 ± 11.3	−10.5 ± 1.8	56 ± 4.6	−10.6 ± 0.8	13 ± 1.0
	200	**−10** ± 1.4	108 ± 7.7	−31 ± 2.9	106 ± 11.1	−105 ± 12.4	33 ± 5.5
**SDVRP**	20	**−6.8** ± 0.96	26 ± 2.0	−7.3 ± 0.44	25 ± 4.8	−7.6 ± 1.48	5 ± 0.86
	50	**−11.7** ± 0.8	56 ± 7.0	−13.1 ± 0.7	53 ± 9.9	−13.1 ± 1.2	11 ± 1.6
	100	**−18** ± 3.5	103 ± 17.1	−25 ± 4.8	100 ± 18.4	−20 ± 2.8	22 ± 4.1
**SMTWTP**	10	**−1.25** ± 0.14	11 ± 1.0	−1.31 ± 0.23	9 ± 0.93	−1.27 ± 0.12	2 ± 0.26
	20	**−2.72** ± 0.17	18 ± 3.6	−2.95 ± 0.49	14 ± 1.1	−2.83 ± 0.14	3 ± 0.52
	50	**−19** ± 3.1	33 ± 2.0	−85 ± 1.2	29 ± 2.0	**−19** ± 3.4	17 ± 2.4
**Task**	**N**	**AM**		**NeuOpt**		**NN**	
		R (±std)	T (±std)	R (±std)	T (±std)	R (±std)	T (±std)
**TSP**	50	**−5.5** ± 0.32	8 ± 1.44	−5.5 ± 0.2	40 ± 3.5	−5.6 ± 0.78	0.50 ± 0.078
	100	−10 ± 1.0	16 ± 2.1	−10.2 ± 0.6	105 ± 7.2	−10.2 ± 1.2	0.50 ± 0.047
	200	−25 ± 2.0	38 ± 4.8	−22 ± 1.2	243 ± 9.4	−12 ± 1.7	1 ± 0.06
**SDVRP**	20	−7 ± 0.67	7 ± 0.45	−6.5 ± 0.5	43 ± 2.5	−7.2 ± 1.10	0.50 ± 0.058
	50	−12.5 ± 1.2	14 ± 1.8	−12.5 ± 0.8	112 ± 5	−11.9 ± 1.6	0.50 ± 0.039
	100	−19 ± 1.2	25 ± 2.0	−17 ± 0.8	265 ± 3.2	−19.5 ± 1.1	1 ± 0.10
**SMTWTP**	10	**−1.25** ± 0.09	3 ± 0.51	−1.25 ± 0.8	18 ± 1.0	-	-
	20	**−2.72** ± 0.42	5 ± 0.80	−3.2 ± 0.3	39 ± 2.5	-	-
	50	−22 ± 0.4	20 ± 1.2	−21.5 ± 0.3	65 ± 4.6	-	-

### 5.7. Ablation Study

#### 5.7.1. IB Objectives

To better understand the contribution of each component of the proposed IBE framework, we conducted an ablation study focusing on the two information bottleneck (IB) objectives. As illustrated in Algorithm 1, IBE introduces two coefficients, $\beta_{1}$ and $\beta_{2}$, to control the strength of the state representation bottleneck and the policy bottleneck, respectively. Specifically, $\beta_{1}$ determines the degree of information compression in the attention-based encoder, encouraging the agent to learn compact and task-relevant representations. Meanwhile, $\beta_{2}$ regulates the exploration–exploitation trade-off in the policy by controlling the regularization strength applied during policy optimization.

We systematically varied these coefficients and evaluated their impact on performance in all three environments (see Table 5). When both IB objectives are disabled ($\beta_{1}=\beta_{2}=0$), the agent effectively reduces to a standard attention-based RL model without information constraints. In this case, the performance significantly deteriorates, confirming that the inclusion of the IB objectives plays a crucial role in stabilizing learning and improving generalization.

On the other hand, setting excessively large coefficients ($\beta_{1}=\beta_{2}=1$) leads to severe performance degradation or even collapse. This is because overly strong compression in the latent representation removes useful task information, while excessive policy regularization hinders effective learning and reduces exploration efficiency. These results highlight the importance of balancing compression and expressiveness: moderate information constraints help prevent overfitting and encourage robust representations, but excessive constraints can suppress critical information flow.

Empirically, we find that setting both coefficients to small values ($\beta_{1}=\beta_{2}=0.01$) provides the best trade-off between stability and performance across all tasks, as reported in Section 5.4, Section 5.5 and Section 5.6.

**Table 5 sensors-25-07572-t005:** Performance comparison across different tasks (TSP, SDVRP, and SMTWTP) for various problem sizes (*N*) and $\left(\beta_{1},\beta_{2}\right)$ pairs. Overall, the combination of small values for $\beta_{1}$ and $\beta_{2}$ yields the best performance across all tasks. Results are averaged over 3 independent runs.

Task	*N*	(β_1_, β_2_) Pairs					
		(0, 0)	(1, 1)	(0.01, 0.01)	(0.5, 0.5)	(0, 1)	(1, 0)
TSP	50	−6.7 ± 0.8	−7.3 ± 0.4	**−5.5 ± 0.58**	−7.1 ± 0.6	−7.5 ± 0.5	−7.9 ± 1.0
	100	−10.5 ± 1.2	−11.2 ± 1.6	**−9 ± 0.48**	−9.4 ± 0.6	−10.6 ± 1.0	−11.1 ± 0.6
	200	−37.1± 2.7	−42.4 ± 4.3	**−10 ± 1.4**	−30.2 ± 1.6	−35.6 ± 2.5	−33.2 ± 3.8
SDVRP	20	−7.2 ± 0.8	−8.1 ± 0.6	**−6.8 ± 0.96**	−7.8 ± 0.8	−8.0 ± 1.0	−8.4 ± 0.4
	50	−12.4 ± 1.6	−13.5 ± 1.6	−11.7 ± 0.8	**−10.9 ± 0.9**	−12.1 ± 0.8	−12.9 ± 1.9
	100	−21.5 ± 1.8	−18.2 ± 1.1	**−18 ± 3.5**	−24.5 ± 1.5	−19.8 ± 2.7	−28.0 ± 3.1
SMTWTP	10	−1.8 ± 0.2	−1.5 ± 0.2	**−1.25 ± 0.14**	−2.0 ± 0.2	−2.8 ± 0.4	−2.1 ± 0.2
	20	−3.1 ± 0.3	−4.4 ± 0.6	**−2.72 ± 0.17**	−3.8 ± 0.4	−5.9 ± 0.8	−4.2 ± 0.5
	50	−22.4 ± 2.7	−19.9 ± 1.1	**−19 ± 3.1**	−24.1 ± 1.8	−23.5 ± 1.9	−24 ± 1.4

#### 5.7.2. Attention Encoder

A critical component of the proposed IBE framework is the attention-based encoder, which serves as the primary mechanism for extracting compact, task-relevant state representations. Unlike conventional feed-forward networks, the attention mechanism explicitly models pairwise dependencies between elements in the input, allowing the agent to reason about global structural relationships, such as spatial proximity among nodes in routing problems or temporal dependencies in scheduling tasks.

To assess the contribution of the attention encoder, we compared IBE using attention against variants employing different numbers of fully connected (FC) layers in their encoders under identical training conditions. The results in Table 6 report performance on TSP tasks of different sizes. Across all scales, the attention-based encoder significantly outperforms its FC counterpart, and the gap widens as problem size increases, illustrating that attention becomes increasingly critical for scalability and generalization.

Among the different numbers of layers, a three-layer FC structure works the best, indicating that although deeper networks have more capacity for expression, they also take more time and data to train, resulting in poor performance within the given budget. We elaborate the comparison with the three-layer FC structure below. For smaller TSP instances (N=50), IBE with attention achieves a final reward of −5.5, compared to −8.5 for the FC encoder. As the problem size grows to N=100 and N=200, the gap expands sharply (−9 vs. −12.2 and −10 vs. −32.1), indicating that feed-forward architectures fail to capture complex relational dependencies among nodes. These findings confirm that attention mechanisms provide a strong inductive bias for learning structured representations, leading to more effective information compression under the state bottleneck objective.

## 6. Conclusions and Future Work

In this work, we proposed IBE, an Information Bottleneck-Enhanced RL method that integrates the information bottleneck principle with attention-based reinforcement learning to improve representation learning, exploration, and training stability in combinatorial optimization tasks. IBE introduces two complementary regularizers: a state representation bottleneck, which enforces compact and task-relevant embeddings, and a policy bottleneck, which encourages information-efficient exploration by constraining the mutual information between states and actions.

Extensive experiments on two routing tasks and one scheduling task demonstrate that IBE consistently outperforms PPO, REINFORCE, AM, and NeuOpt in convergence speed, performance, and stability across multiple problem scales. Ablation studies confirm that both bottlenecks and the attention encoder contribute synergistically to improved performance.

In our work, the coefficients of two IB objectives are fixed. It would be interesting to explore adaptive scheduling of the these coefficients to dynamically balance compression and exploration. Furthermore, extending IBE to more complex constraints and larger-scale industrial instances is also a promising direction.

## Figures and Tables

**Figure 1 sensors-25-07572-f001:**
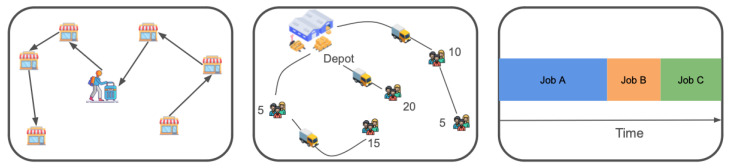
The three tasks we used in our work. From left to right, they are a Traveling Salesman Problem, Split-Delivery Vehicle Routing Problem, and Single-Machine Total Weighted Tardiness Problem.

**Figure 2 sensors-25-07572-f002:**
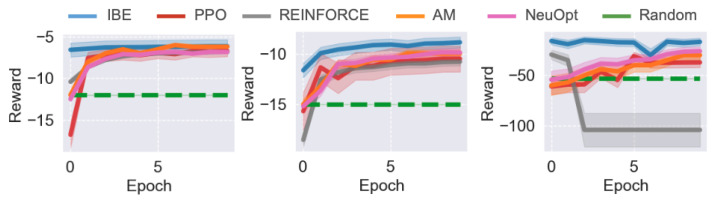
Results of different agents on different TSP tasks. From (**left**) to (**right**), N=50,100,200. Results are averaged over 3 independent runs and standard deviations are reported.

**Figure 3 sensors-25-07572-f003:**
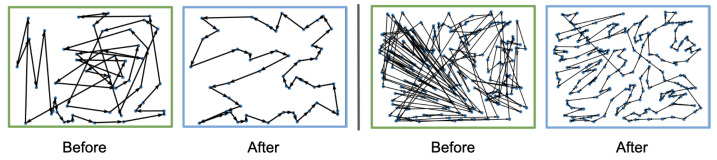
Visualization of solutions at the start (green) and end (blue) of training in different TSP tasks. (**Left**): N=50; (**right**): N=200.

**Figure 4 sensors-25-07572-f004:**
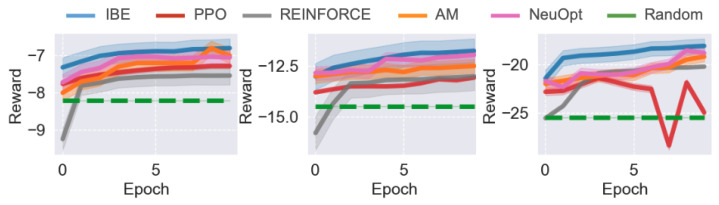
Results of different agents on different SDVRP tasks. From (**left**) to (**right**), N=20,50,100. Results are averaged over 3 independent runs and standard deviations are reported.

**Figure 5 sensors-25-07572-f005:**
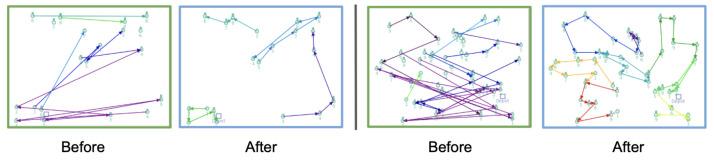
Visualization of solutions at the start (green) and end (blue) of training in different SDVR tasks. (**Left**): N=20; (**right**): N=50.

**Figure 6 sensors-25-07572-f006:**
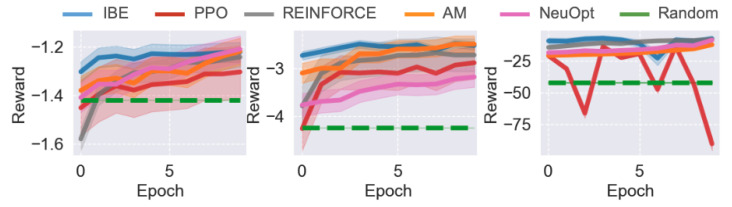
Results of different agents on different SMTWTP tasks. From (**left**) to (**right**), N=10,20,50. Results are averaged over 3 independent runs and standard deviations are reported.

**Table 1 sensors-25-07572-t001:** State space of the TSP environment.

Attribute	Description
locs	2D coordinates of all cities/customers.
first_node	Index of the starting node (first city visited)
current_node	Index of current city the agent is in
i	Step counter (number of cities visited so far)
action_mask	Indicates which cities are still available to visit (1 = True means available, 0 = False means already visited)

**Table 2 sensors-25-07572-t002:** State space of the SDVRP environment.

Attribute	Description
depot_location	The coordinates of the central depot where the vehicle starts and returns.
customers_location	The coordinates of all customers to be served.
customer_demand	The total or remaining demand associated with each customer.
current_location	The current position of the vehicle in the routing space.
remaining_capacity	The available load capacity of the vehicle.

**Table 3 sensors-25-07572-t003:** State space of the SMTWTP environment.

Attribute	Description
job_due_time	The due date (deadline) for each job, indicating when the job should ideally be completed.
job_weight	The importance or priority of each job, used to calculate weighted tardiness.
job_process_time	The processing time required to complete each job.
current_node	The index or identifier of the currently selected job or state in the scheduling sequence.
action_mask	A binary mask indicating which jobs are still available for scheduling (1 = available, 0 = unavailable).
current_time	The cumulative time elapsed so far, representing the completion time of the last scheduled job.

**Table 6 sensors-25-07572-t006:** Performance of IBE with attention encoder and different FC encoders in TSP tasks. We see that IBE with attention encoders yields the best performance. Results are averaged over 3 independent runs.

N	Attention	3-Layer FC	5-Layer FC	7-Layer FC
50	−5.5±0.58	−8.5±0.9	−8.7±2.1	−9.7±1.4
100	−9±0.48	−12.2±1.3	−15.3±1.1	−18.4±2.8
200	−10±3.1	−32.1±5.1	−36.3±2.7	−42.2±5.7

## Data Availability

The data used in this study were generated using publicly available benchmark environments for combinatorial optimization, including the Traveling Salesman Problem, Split-Delivery Vehicle Routing Problem, and Single-Machine Total Weighted Tardiness Problem. All data are synthetic and can be reproduced by following the experimental setup and parameter settings described in the paper. The code used to generate the data and conduct the experiments is available from the corresponding authors upon reasonable request.

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
