# Peer review of "Information Bottleneck-Enhanced Reinforcement Learning for Solving Operation Research Problems"

_sensors, 2025, doi:10.3390/s25247572_

Round 1
Reviewer 1 Report
Comments and Suggestions for Authors
(can also be seen in the attachment)
Review Comments
- Strengthen Literature Review
The following paper related to reinforcement learning should be reviewed:
Zhang, J., Li, X., Yuan, Y., Yang, D., Xu, P. and Au, F.T., 2024. A multi-agent ranking proximal policy optimization framework for bridge network life-cycle maintenance decision-making. Structural and Multidisciplinary Optimization, 67(11), p.194.
Zhang, Y., Zhao, W., Wang, J. and Yuan, Y., 2024. Recent progress, challenges and future prospects of applied deep reinforcement learning: A practical perspective in path planning. Neurocomputing, 608, p.128423.
Chen, X., Liu, S., Zhao, J., Wu, H., Xian, J. and Montewka, J., 2024. Autonomous port management based AGV path planning and optimization via an ensemble reinforcement learning framework. Ocean & Coastal Management, 251, p.107087. - 2. Complete the Algorithmic Description
Algorithm 1 currently shows only the actor update, but PPO requires both actor and critic components. Please add the critic loss function, the critic update step, clarification on whether and how the bottleneck objectives affect the critic, and details on how value targets and advantage estimates are computed. Without these details, the method is incompletely specified and difficult to reproduce. - 3. Provide Complete Technical Details
(1) For the state bottleneck, specify: the exact form of pθ(T|S) including mean parameterization, whether σ is fixed or learned, diagonal or full covariance structure, the closed-form KL divergence formula used.
(2) Clarify if the encoder is shared between actor and critic.
(3) For the policy bottleneck with masked actions, explain how the prior p(A) is defined over the feasible action set only (renormalized after masking) and how infeasible actions are handled in the logits/softmax and KL computation. - 4. Provide more detailed hyperparameter setting and results
Add: instance generation details (distributions, scaling, normalization), hardware specifications and wall-clock time, all PPO hyperparameters (GAE, clipping ε, batch size, update epochs, entropy coefficients, learning rates), mean ± standard deviation over multiple random seeds with statistical significance tests. - 5. Fix Inconsistencies and Presentation Issues
Correct the following:
(1) "Travel Sales Man" → "Traveling Salesman Problem";
(2) IBE vs. IBE-RL: double check whether mistaken using IBE instead of IBE-RL in text;
(3) ensure all figures have properly labeled axes with units and consistent scales across subfigures.
(4) double check other grammar problems

Author Response
We sincerely appreciate the valuable comments from the reviewer. Please find in the attached document our reply.

Reviewer 2 Report
Comments and Suggestions for Authors
Please refer to the attached pdf.

Author Response

(The authors gave the same response as above.)

Round 2
Reviewer 2 Report
Comments and Suggestions for Authors
Minor comments
- Language and minor wording
- The language is much improved overall. Please do one more light pass (or copy-editing) to remove residual minor issues and ensure consistent, polished phrasing (e.g., “travel sales man task”, “Travel Sales Man Problem”, “TSP task”, etc.).
- Small clarifications
- When you discuss the policy bottleneck vs entropy regularization, you might add one concrete sentence summarizing how a non-uniform prior p(A) could encode domain knowledge (for example, prioritizing feasible or shorter routes), further emphasizing the practical value of the framework.
- In the limitations/future work, consider briefly mentioning potential extensions to more complex constraints or larger-scale industrial instances.
Author Response
We thank the reviewer for valuable comments. Please find the attached document of our reply to your concerns.
